# Cellular Metabolism and Bioenergetic Function in Human Fibroblasts and Preadipocytes of Type 2 Familial Partial Lipodystrophy

**DOI:** 10.3390/ijms23158659

**Published:** 2022-08-04

**Authors:** Cristina Algieri, Chiara Bernardini, Fabiana Trombetti, Elisa Schena, Augusta Zannoni, Monica Forni, Salvatore Nesci

**Affiliations:** 1Department of Veterinary Medical Sciences, University of Bologna, 40064 Ozzano Emilia, Italy; 2CNR Institute of Molecular Genetics “Luigi Luca Cavalli-Sforza”, Unit of Bologna, 40126 Bologna, Italy; 3IRCCS Istituto Ortopedico Rizzoli, 40136 Bologna, Italy; 4Health Sciences and Technologies-Interdepartmental Center for Industrial Research (CIRI-SDV), Alma Mater Studiorum, University of Bologna, 40126 Bologna, Italy

**Keywords:** mitochondrial oxidative metabolism, glycolysis, type-2 familial partial lipodystrophy, ATP production, fibroblasts, pre-adipocytes

## Abstract

LMNA mutation is associated with type-2 familial partial lipodystrophy (*FPLD2*). The disease causes a disorder characterized by anomalous accumulation of body fat in humans. The dysfunction at the molecular level is triggered by a lamin A/C mutation, impairing the cell metabolism. In human fibroblasts and preadipocytes, a trend for ATP production, mainly supported by mitochondrial oxidative metabolism, is detected. Moreover, primary cell lines with *FPLD2* mutation decrease the mitochondrial ATP production if compared with the *control*, even if no differences are observed in the oxygen consumption rate of bioenergetic parameters (i.e., basal and maximal respiration, spare respiratory capacity, and ATP turnover). Conversely, glycolysis is only inhibited in *FPLD2* fibroblast cell lines. We notice that the amount of ATP produced in the fibroblasts is higher than in the preadipocytes, and likewise in the *control*, with respect to *FPLD2*, due to a more active oxidative phosphorylation (OXPHOS) and glycolysis. Moreover, the proton leak parameter, which characterizes the transformation of white adipose tissue to brown/beige adipose tissue, is unaffected by *FPLD2* mutation. The metabolic profile of fibroblasts and preadipocytes is confirmed by the ability of these cell lines to increase the metabolic potential of both OXPHOS and glycolysis under energy required independently by the *FPLD2* mutation.

## 1. Introduction

Laminopathies are several diseases arising from mutations in the *LMNA* gene. Heterozygous mutations of the *LMNA* gene encoding lamin A/C, the main nuclear intermediate filament proteins of the nuclear envelope, are responsible for complex diseases that can combine lipodystrophy, metabolic complications, osteoporosis and osteolysis, and signs of accelerated ageing [1,2,3]. Type 2 Familial Partial Lipodystrophy (*FPLD2*, Online Mendelian Inheritance in Man (OMIM) code—#151660) is lipodystrophy that induces subcutaneous adipose tissue loss in the limbs and buttocks from puberty, but augments the fat accumulation in the neck and face. As a consequence, people have a predisposition to insulin resistance, high blood pressure, liver steatosis, and an increased risk of cardiovascular diseases [4]. Brown adipose tissue (BAT)/white adipose tissue (WAT) interchange is impaired by adipogenesis disorder, deregulation of the adipocyte lipid droplets, and caveolae within adipocytes. These adipose tissue dysfunctions can lead to lipodystrophy pathogenesis [5].

The mitochondria support several important functions in adipose tissue, including the production of ATP, the oxidation of fatty acids, and the balance of triglycerides in the cell. In addition to their involvement in diseases associated with energy deficiency, mitochondrial dysfunctions cause harmful effects on adipocyte differentiation, lipid metabolism, insulin sensitivity, and oxidative capacity on thermogenesis, which consequently leads to metabolic diseases [6]. For instance, it is known that in adipocytes, when mitochondrial activity is impaired, a reduced β-oxidation of fatty acids occurs, and this causes a higher content of cytosolic free fatty acids, also altering the absorption of glucose [7,8]. HIV patients treated with highly active antiretroviral therapy, which has as a side effect of the onset of lipodystrophy associated with peripheral lipoatrophy and an increase in the amount of visceral WAT, have reduced mitochondrial activities [9,10]. Indeed, it was reported that the drugs used in this therapy might induce mitochondrial damage, such as the inhibition of crucial mitochondrial enzymes (i.e., enzymes involved in lipid metabolism or the damage induced to mtDNA), leading to lipodystrophy. Similarly, in multiple symmetric lipomatosis, the impairment of the mitochondria plays a crucial role [11].

Mitochondriogenesis and the expression of uncoupling protein 1 (UCP1) in the BAT activation are stimulated by exposure to temperatures below thermoneutrality and food intake [12]. Moreover, UCP1 expression characterizes BAT formation under adrenergic signalling mediated by β_3_-adrenoreceptors. This biological event in adipose tissue involves the adipocyte lipid regulation by autophagy and the mitochondrial degradation pathways [13]. Indeed, *FPLD2* preadipocytes exhibit an intense vesicle accumulation. The involvement of a mitophagic alteration in *FPLD2* cells is also related to transcription factor Oct-1 in regulating the mTOR signalling pathway, which might affect adipocyte determination and differentiation [14]. Moreover, *FPLD2* patients might have a BAT functional failure. Even if fat from the neck area of *FPLD2* patients is ascribable to WAT, the UCP1 expression in adipose tissue from that district of the body has been ascertained by highlighting the BAT origin [15]. Therefore, the adipose tissue accumulation in the neck of *FPLD2* patients is morphologically identifiable as WAT, even if is biochemically characterized as BAT. In all likelihood, non-programmed autophagic activation that alters BAT and WAT turnover in *FPLD2* patients is consistent with morphological and biochemical results [16]. Further, selective adipose tissue knockout of mitochondrial transcription factor A in mice has been shown to result in lipodystrophy and insulin resistance [17], suggesting that mitochondrial dysfunction might induce lipodystrophy syndrome.

Fibroblast cultures obtained from low-invasive biopsies can generate other cell types, among them either adipocytes and preadipocytes. The correct metabolic function of the adipose tissue might depend on precursor cell differentiation capacity [18].

Through a comparative analysis of metabolic pathways in fibroblasts and preadipocytes, involved in the onset of *FPLD2*, this study aims to highlight for the first time the oxidative aspects and anaerobic glycolysis metabolism that support *FPLD2* upstream of the differentiation process. By considering the prominent mitochondrial role in guaranteeing correct cellular function in energy production, the results allow us to shed light on molecular mechanisms that can be used as possible targets to counteract *FPLD2* alterations.

## 2. Results

### 2.1. ATP Production and Bioenergetic Parameters of Fibroblasts and Preadipocytes FPLD2

The cellular ATP synthesis, namely ATP production rate, related to the conversion of glucose to lactate in the glycolytic pathway (glycoATP production rate) and to oxygen consumption in mitochondrial oxidative phosphorylation (OXPHOS) (mitoATP production rate) has been characterized. We performed the ATP production rate analysis on wild-type (named as *control*) or *LMNA* mutated fibroblasts and preadipocytes (named as *FPLD2*) by injecting oligomycin, to inhibit mitochondrial ATP synthesis, and then rotenone plus antimycin A to block mitochondrial respiration (Figure 1A). The results highlighted the feature of a cell metabolism based on substrate oxidation.

MitoATP production in *control* fibroblasts was twice as high as *FPLD2*, whereas in preadipocytes, the mitoATP production in the *control* was 1.5 times greater than *FPLD2*. Only glycoATP production in the *control* fibroblasts had a significant activation—four times greater than that of other cell lines. Consistently, the total ATP production of *FPLD2* fibroblasts and preadipocytes was inhibited by 46% and 19% compared to *control* fibroblasts and preadipocytes, respectively. Moreover, the total ATP production of *control* and *FPLD2* fibroblasts was about 150% and 65% higher than *control* and *FPLD2* preadipocytes, respectively, due to a consequent rise in mitochondrial ATP synthesis.

The ratio between the mitoATP production rate and glycoATP production rate (ATP rate index) is the metric for detecting changes and/or differences in the metabolic phenotype. An ATP rate index > 1 means a more oxidative phenotype (OXPHOS pathway); conversely, a ratio < 1 means a more glycolytic and less oxidative phenotype. Even if *FPLD2*, which is caused by an autosomal dominant mutation in the *LMNA* gene, could modify ATP production, the ratio between mitoATP vs. glycoATP production rates was always >1 (Figure 1B). In detail, *control* fibroblasts increased the propensity to produce glycoATP with respect to the other cell lines, but cell metabolism relied mainly on substrate mitochondrial oxidation for ATP production.

The cell respiration profile of *FPLD2* and *control* fibroblasts (Fibro *FPLD2* and Fibro *control*) and preadipocytes (Pre-adip *FPLD2* and Pre-adip *control*) was analyzed (Figure 1C). The key parameters of cell respiration obtained were (i) basal respiration detected as baseline oxygen consumption rate (OCR) before oligomycin (olig) addition; (ii) minimal respiration measured as OCR in the presence of oligomycin; (iii) maximal respiration evaluated as OCR after carbonyl-cyanide-4-(trifluoromethoxy) phenylhydrazone (FCCP) addition; (iv) non-mitochondrial respiration evaluated as OCR in the presence of rotenone plus antimycin A (rot + AA), the respiratory chain inhibitors. The latter was subtracted from all the mitochondrial parameters. We noticed no difference in mitochondrial respiration between *FPLD2* and the *control* in all bioenergetics parameters of both cell lines analyzed (Figure 1D). The results obtained from OCR values showed that basal respiration was higher in fibroblasts than in preadipocytes. In general, the basal respiration of cells is strongly controlled by ATP turnover and proton leak. The basal respiration inhibited by oligomycin (the specific inhibitor of ATP synthase) corresponds to the ATP turnover, which is the rate of mitochondrial ATP synthesis in a defined basal state, whereas the residual activity, or oligomycin-insensitive respiration, is the proton leak [19]. Therefore, the proton leak is the basal respiration subtracted from the respiration sensitive to oligomycin (ATP turnover) and indicates the re-entry of H^+^ in the intermembrane space, bypassing the F_1_F_O_-ATP synthase activity [20]. In all the cell lines tested, the so-called proton leak attributed to the UCPs activity was not affected (Figure 1D). From the difference between basal respiration and minimal respiration (OCR in presence of oligomycin) arises the ATP turnover, or oligomycin-sensitive respiration, used for ATP synthesis [19]. We revealed statistically significant differences in the ATP turnover between fibroblasts and preadipocytes. The higher ATP synthesis of fibroblasts than preadipocytes was independent of the *FPLD2* condition.

The coupling efficiency, known as basal mitochondrial oxygen consumption used for ATP synthesis, has the maximal value of 1.0 a.u. and is only obtained when all the basal respiration is sensitive to oligomycin. As an empirical value, it is obtained as ATP turnover/basal respiration ratio. In *FPLD2* and *control* fibroblasts and *control* preadipocytes, the coupling efficiency was 0.80 ± 0.04, 0.76 ± 0.03, and 0.88 ± 0.02 a.u., respectively, whereas *FPLD2* preadipocytes had a significantly lower value (0.67 ± 0.04 a.u.). The OCR in presence of FCCP (an uncoupling agent that produced the maximal cell respiration) showed no difference in the same cell line, i.e., fibroblasts or preadipocytes. Contrastingly, the maximal respiration of *FPLD2* and *control* fibroblasts was markedly increased compared to the *FPLD2* and *control* preadipocytes. The difference between the maximal and the basal respiration provided the spare capacity, which represents the ability of cells to respond to an increased energy demand under conditions of physiological or pathophysiological stimulus and can be considered as a measure of the flexibility of the oxidative phosphorylation machinery in tracing the transition of cell fate and survival [21]. In other words, it is the ability of the substrate supply and the electron transport chain to respond to an increase in energy demand [19]. For this parameter, we noticed a trend similar to that encountered for maximal respiration and ATP turnover.

### 2.2. Glycolytic Rate of Fibroblasts and Preadipocytes FPLD2

By the proton efflux rate (PER), we identified the amount of lactate produced during glycolysis as the rate of protons extruded to the extracellular medium. The calculated rates account for the contribution of CO_2_ to extracellular acidification derived from Krebs cycle activity in the total PER and glycolysis alone in the glycoPER (Figure 2A,B). The results have allowed for real-time measurements of changes in glycolysis rates. Subtraction of mitochondrial acidification from the total proton efflux rate resulted in the glycolytic proton efflux rate (glycoPER). The basal glycolysis was markedly elevated in *control* fibroblasts in comparison with the other cell lines (Figure 2C). Additionally, we noticed that the compensatory glycolysis of the preadipocytes, which described the glycolysis capacity to meet the cells’ energy demands with inhibited oxidative phosphorylation, was lower than that of the fibroblasts. From the real-time extra cellular acidification rate (ECAR) and OCR measurement, the *FPLD2* fibroblasts showed a significant decrease in both glycoPER activities compared to *control* (Figure 2C). The contribution of mitochondria/CO_2_ to extracellular acidification was 60% in *FPLD2* fibroblasts, whereas it was less than 40% in *control* fibroblasts, and 35% for both *FPLD2* and *control* preadipocytes (Figure 2D).

### 2.3. Metabolic Phenogram

We have measured mitochondrial respiration and glycolysis under baseline and stressed conditions (i.e., under an induced energy demand and specifically in the presence of stressor compounds such as oligomycin plus FCCP) by evaluating two main parameters of cell energy metabolism known as metabolic phenotypes (baseline and stressed phenotype) (Figure 3A) and metabolic potential (Figure 3B). The baseline was obtained from the OCR and ECAR values in cells under the starting condition in the presence of substrates (10 mM glucose, 1 mM pyruvate, and 2 mM glutamine). The cell lines analyzed depicted an energetic profile, with a proportional increase in aerobic and glycolytic metabolism (Figure 3A). The increment of energy production via cell respiration and glycolysis was defined by the cellular metabolic potential as the % increase of the stressed phenotype over the baseline phenotype of OCR and ECAR to meet the cell energy demand (Figure 3B). The stressed OCR was about 100% higher than the baseline OCR. There was no significant difference in the metabolic potential in the presence or absence of mutation or between fibroblasts and preadipocytes. In comparison with the baseline phenotype, the metabolic potential of ECAR in *FPLD2* fibroblasts or *FPLD2* and *control* preadipocytes increased by more than 150%, whereas by only 100% in *control* fibroblasts (Figure 3B).

## 3. Discussion

The results showed that *FPLD2* and *control* fibroblasts or preadipocytes mainly relied on mitochondrial oxidative metabolism as the main pathway to produce ATP. In general, by the mitochondria oxidation of substrates, the OXPHOS accounts for the production of more than 80% of cellular ATP by highlighting the cell’s ability to synthesize more ATP molecules at a lower bioenergetic cost [22]. The ATP rate index outlines that fibroblasts and preadipocytes switched to more aerobic metabolism, namely OXPHOS vs. glycolysis, even if the *control* fibroblasts had a lower ATP rate index than the other cell lines. Therefore, the results strengthen the role of mitochondrial OXPHOS, which is the leading energy supplier for *FPLD2* and *control* fibroblasts or preadipocytes cell growth. Pathological conditions are often associated with impaired mitochondrial bioenergetics [23]. However, the *LMNA* gene mutation affected cell respiration, whereas there were no highlighted differences in mitochondrial parameters between the cell lines. Importantly, the mutation decreases the amount of mitoATP production. Since fibroblasts and preadipocytes relied on mitochondria to support the main ATP production, as a consequence, the total ATP production was also less in *FPLD2* than in the *control*. The adipocytes of brown/beige adipose tissue are characterized by the presence of UCP1. Consistently, adipocytes’ mitochondrial proton leak activity, which is joined to the dissipation of the proton motivating force of the protein UCP family, can generate heat during mitochondrial respiration. The mutation might invalidate the dissipation of the mitochondrial proton gradient, and this phenomenon might be linked to lipodystrophy [16]. On the other hand, cellular energy metabolism, in turn, influences the dynamics of mitochondrial morphology. We can assert that the *FPLD2* negatively alters the physiological mitochondrial cycle of either fusion or fission, which causes the deterioration of mitochondrial energy production [24].

The mutation of the *LMNA* gene had an inhibitory effect on glycolysis in fibroblasts, but not in preadipocytes. We evaluated the glycolysis rates for basal conditions and compensatory glycolysis following mitochondrial inhibition. Thus, we provided an accurate measurement of energy production through the extracellular acidification generated by the glycolic pathway by ruling out the Krebs cycle-derived CO_2_ during mitochondrial oxidative metabolism. Indeed, when shutting down the oxidative phosphorylation, *FPLD2* fibroblasts could not drive the cell to use glycolysis to satisfy the energy demand of the cells. Only the fibroblast results in Figure 2C highlighted the negative effect of mutation on the glycolysis pathway. Most cells possess the ability to reverse between these two pathways, thereby adapting to changes in environmental conditions or cellular function activations [25].

In addition to this, the simultaneous measurement of the two major energy-producing pathways of the cell, i.e., mitochondrial respiration and glycolysis, under basal and induced energy demand, depicted the metabolic phenotype of *FPLD2* and *control* fibroblasts or *FPLD2* and *control* preadipocytes that switched from a quiescent to an energetic profile. The depicted quiescent profile increased under stressed conditions. The stressed phenotype was obtained by the OCR and ECAR values in cells after the addition of stressor compounds: oligomycin plus FCCP. Indeed, oligomycin inhibited mitochondrial ATP production, blocking the F_1_F_O_-ATPase [26]. The cells compensated for the failed oxidative phosphorylation by increasing the glycolysis rate. At the same time, when the electrochemical gradient of H^+^ in the mitochondria was dissipated by the ionophore FCCP, the uncoupled respiration drove the highest oxygen consumption in mitochondria. This energy transition by OXPHOS-glycolysis exchange was corroborated by the increase in metabolic potential, which relied on the balanced activation of mitochondrial respiration and glycolysis.

*FPLD2* affected the energy metabolism of fibroblasts and preadipocytes by reducing the amount of energy produced mainly at the mitochondrial level. Importantly, we did not detect a decrease in mitochondrial bioenergetic parameters, and we might assert that the decreased ATP production might be related to the lower quantity or number of active mitochondria in the cell. Conversely, only in fibroblasts, *FPLD2* can inhibit glycolysis. A common feature between fibroblasts and preadipocytes was the propensity to increase their cellular metabolism when greater energy production was required. Consistently, considering the cell metabolism results, we may argue that *FPLD2* impaired mitochondrial homeostasis (biogenesis/mitophagy) and dynamics (fusion/fission) [27].

A reduced synthesis of mitochondrial ATP, associated with unaltered specific mitochondrial parameters, makes us think that the molecular mechanisms underlying *FPLD2* do not involve specific enzymes of mitochondrial oxidative metabolism. Consistently, it could instead alter particular signalling pathways, such as the PTEN-induced putative kinase 1 (PINK1)-Parkin signalling pathway or the mitophagic receptors Bnip3 and Nix [28,29,30]. This would explain the reduction in the number of active mitochondria. Indeed, it is known that the dysregulation of the mitophagy process can contribute to the onset of degenerative diseases, as well as support the ageing process [31].

## 4. Materials and Methods

### 4.1. Chemicals

Seahorse XF Assay Kits and Reagents were purchased from Agilent. All other chemicals were reagent grade and used without purification. Quartz double-distilled water was used for all reagent solutions, except when differently stated. All cell materials were purchased from Thermo Fisher Scientific (Waltham, MA, USA), and all plastic supports for cellular culture were purchased from BD-Falcon (Corning, NY, USA).

### 4.2. Cell Culture

Primary cell lines of fibroblasts and preadipocytes were provided by BioLaM bio-bank (CNR Institute of Molecular Genetics “Luigi Luca Cavalli-Sforza”, Bologna, Italy). Cells were cultured in Dulbecco’s Modified Eagle Medium (DMEM) (4.5 g/L glucose) and added with 20% of fetal bovine serum, 1 × antibiotic–antimycotic solution in a 5% CO_2_ atmosphere at 37 °C and were split weekly. For the cryopreservation, cells were suspended in 1 mL of freezing medium (70% FBS, 20% complete medium and 5% DMSO). An inverted Eclipse Microscope (TS100) (Leica Microsystems, Milan, Italy) with a digital C-Mount Nikon photo camera (TP3100) (Nikon, Amstelveen, The Netherlands) was used to check cell morphology.

### 4.3. Cellular Metabolism

The simultaneous measurement of OCR, an index of cell respiration (pmoL/min), and ECAR, an index of glycolysis (mpH/min), was performed by the Seahorse XFp analyzer (Agilent). A total of 20 k/well of fibroblasts and 10 k/well of preadipocytes cells were grown in XFp cell culture mini plates (Agilent, Santa Clara, CA, USA) for 24 h. On the experiment day, the cell lines were switched to freshly made Seahorse XF DMEM medium pH 7.4 supplied with 10 mM glucose, 1  mM sodium pyruvate, and 2 mM L-glutamine. The plates were incubated at 37 °C in the air for 45′ before measuring OCR and ECAR by the adequate programs (ATP Rate Assay, Cell Mito Stress Test, Glycolytic Rate Assay, and Cell Energy Phenotype Test). The injection ports of XFp sensor cartridges, which were hydrated overnight with XF calibrant at 37 °C, were loaded with 10× concentration of inhibitors according to the instructions provided by the Seahorse XFp ATP Rate Assay, Cell Mito Stress Test, Glycolytic Rate Assay, and Cell Energy Phenotype Test. The final concentration used for ATP Rate Assay were 1.5 μM oligomycin (port A) and 0.5 μM rotenone (Rot) plus antimycin A (AA) (port B). For the Cell Mito Stress Test, the final concentrations were 1.5 μM oligomycin (olig) (port A), 2.0 μM carbonyl-cyanide-4-(trifluoromethoxy) phenylhydrazone (FCCP (port B) with fibroblasts, 3.0 μM FCCP with preadipocytes, and 0.5 μM rotenone plus antimycin A (port C) [32]. In the Glycolytic Rate Assay, we used 0.5 μM rotenone plus antimycin A (rot + AA) (port a) and 50 mM 2-deoxy-D-glucose (2-DG), while for the Cell Energy Phenotype Test, the final concentrations were 1.5 μM oligomycin plus 2.0 μM, or 3.0 μM FCCP (port A) in presence of fibroblasts, preadipocytes, or adipocytes. All the analyses were run at 37 °C. All data were analyzed by WAVE software version 2.6.1. OCR and ECAR values were normalized to the total number of cells per well. All parameter values were calculated per well according to the manufacturer’s instructions. ATP Rate Assay, Mito Stress Test, Glycolytic Rate Assay and Cell Energy Phenotype Test were all carried out three times on different days [33].

### 4.4. Statistical Analysis

Statistical analyses were performed by SIGMASTAT software. Each treatment was replicated three or eight times (viability test) in three independent experiments. Data were analyzed by the Student’s *t*-test, or by one-way analysis of variance (ANOVA) followed by the Student–Newman–Keuls test, when *F* values indicated significance (*p* ≤ 0.05). Percentage data were arcsin-transformed before statistical analyses to ensure normality.

## 5. Conclusions

In addition to multilocular lipid droplets present in BAT and absent in WAT, which could be tightly associated with mitochondria activity [34], a very high number of mitochondria are present in BAT. Mitochondria sustain fatty acid oxidation and glucose uptake [35] and express a high amount of UCP1 that uncouples respiration and dissipates chemical energy as heat. This last difference may not be noticed in a BAT with an altered mitochondrial cycle in which fission and mitophagy prevail over fusion and biogenesis. Accordingly, there would be a low number of mitochondria in BAT of *FPLD2* patients, as well as in WAT, even if the mitochondria express UCP1, the molecular target that characterizes BAT.

Our studies, which were focused on adipocyte precursors, allow us to hypothesize that *FPLD2* induces impaired cell bioenergetics. *FPLD2* might involve, in an unknown way, a dysfunction of the signalling pathways that regulate the processes of mitophagy, reducing the number of active mitochondria. In the mitochondrial ATP deficiency condition of lipodystrophy, these results, considering a possible role of homeostasis and mitochondrial dynamics linked to the *FPLD2* mutation, could allow for the discovery of strategies to combat ageing-associated diseases.

## Figures and Tables

**Figure 1 ijms-23-08659-f001:**
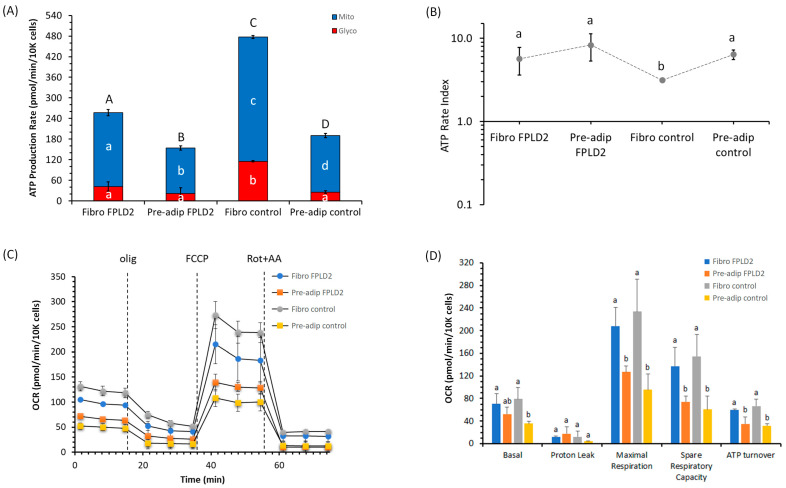
Bioenergetic metabolism in *FPLD2* and *control* fibroblasts (Fibro) or preadipocytes (Pre-adip). (**A**) Evaluation of the real-time ATP production rate by mitochondrial OXPHOS (blue) (█) or by glycolysis (red) (█). Different lower-case letters indicate significantly different values (*p* ≤ 0.05) among cell lines in the same metabolic pathway; different upper-case letters indicate different values (*p* ≤ 0.05) among cell lines in ATP production rates due to the sum of OXPHOS plus glycolysis. (**B**) The ATP rate index, calculated as the ratio between the mitochondrial ATP production rate and the glycolytic ATP production rate, is shown on the *y*-axis (logarithmic scale) in fibroblasts *FPLD2* and *control* and preadipocytes *FPLD2* and *control*. Different lower-case letters indicate significantly different values (*p* ≤ 0.05) among cell lines. (**C**) The mitochondrial respiration profile in cell lines is evaluated as OCR under basal respiration conditions and after the addition of 1.5 μM olig, 2.0 μM, or 3.0 μM FCCP with fibroblasts and preadipocytes, respectively, and a mixture of 0.5 μM rot + AA. Compound injections are shown as dotted lines. (**D**) Mitochondrial parameters (basal respiration, ATP production, proton leak, maximal respiration, spare respiratory capacity, non-mitochondrial oxygen consumption, ATP turnover) in *FPLD2* fibroblasts (blue) (█), in *FPLD2* preadipocytes (orange) (█), in *control* fibroblasts (grey) (█), and *control* preadipocytes (gold) (█). Different letters indicate significant differences (*p* ≤ 0.05) among treatments within the same parameter. Data expressed as column charts ((**A**,**C**) plots) and points ((**B**,**D**) plots), represent the mean ± SD (vertical bars) from three experiments carried out on different cell preparations.

**Figure 2 ijms-23-08659-f002:**
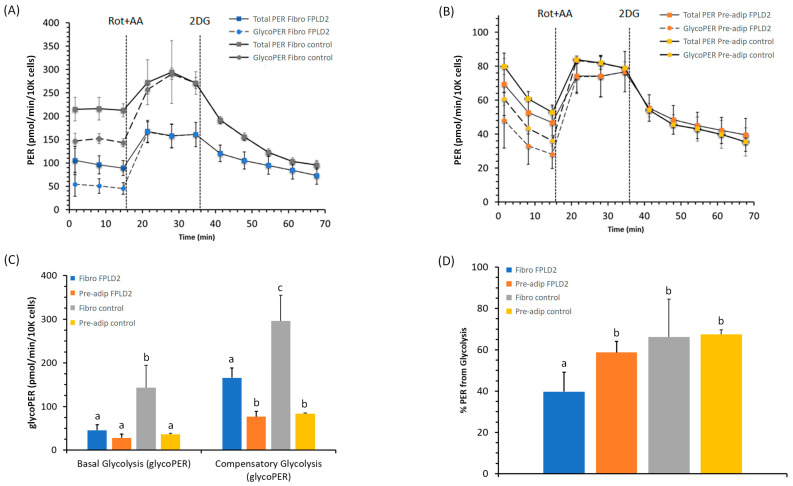
Glycolytic rate assay profile of *FPLD2* and *control* fibroblasts (Fibro) (**A**) or *FPLD2* and *control* preadipocytes (Pre-adip) (**B**). Proton efflux glycolytic (● and dashed line) and total proton efflux, glycolytic plus mitochondrial-derived acidification (■ and solid line). The glycolytic rate profile in cell lines evaluates the proton efflux rate (PER) as a basal glycolysis record before the addition of 0.5 μM rot + AA. The rate of glycolysis after the injection of rot + AA defines the compensatory glycolysis in cells The injection of 50 mM 2-deoxy-D-glucose (2-DG) confirms the extracellular acidification attributed to glycolysis. Compound injections are shown as dotted lines. (**C**) The PER derived from glycolysis parameters (basal glycolysis and compensatory glycolysis) in cell lines and (**D**) glycolytic PER percentages of the Total PER in *FPLD2* fibroblasts (blue) (█), in *FPLD2* preadipocytes (orange) (█), in *control* fibroblasts (grey) (█), and in *control* preadipocytes (gold) (█). Data expressed as points ((**A**,**B**) plots) and column charts ((**C**,**D**) plots), represent the mean ± SD (vertical bars) from three experiments carried out on different cell preparations. Different lowercase letters indicate different values (*p* ≤ 0.05) between cell lines of the same treatment group.

**Figure 3 ijms-23-08659-f003:**
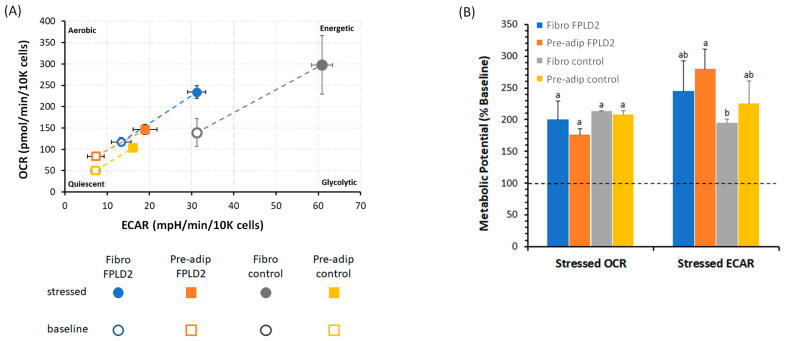
The energy map of *FPLD2* and *control* fibroblasts (Fibro) or *FPLD2* and *control* preadipocytes (Pre-adip). Baseline (empty spheres or squares) and stressed (full spheres or squares) phenotypes of *FPLD2* fibroblasts in (blue sphere), *control* fibroblasts (orange square), *FPLD2* preadipocytes (grey sphere), and *control* preadipocytes (gold square). The metabolic potential in “Stressed OCR” and “Stressed ECAR” is expressed as % “Baseline OCR” and “Baseline ECAR” (dashed horizontal line), in *FPLD2* fibroblasts (blue) (█), in *FPLD2* preadipocytes (orange) (█), in *control* fibroblasts (grey) (█), and in *control* preadipocytes (gold) (█). Data expressed as points (**A**) and column chart (**B**) represent the mean ± SD (vertical and where present, also horizontal bars) from three experiments carried out on different cell preparations. Different letters indicate significant differences (*p* ≤ 0.05) among treatments within the same parameter.

## Data Availability

Not applicable.

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
