# Peer review of "Cellular Metabolism and Bioenergetic Function in Human Fibroblasts and Preadipocytes of Type 2 Familial Partial Lipodystrophy"

_ijms, 2022, doi:10.3390/ijms23158659_

Round 1
Reviewer 1 Report
- Comments are attached.
Author Response
Thank you for giving us the opportunity to submit a revised draft of the manuscript “Cellular metabolism and bioenergetic function in human fi-broblasts and pre-adipocytes of type 2 familial partial lipo-dystrophy” for publication in the Int. J. Mol. Sci. We appreciate the time and effort that you and the reviewers dedicated to providing feedback on our manuscript and are grateful for the insightful comments on and valuable improvements to our paper. We have incorporated most of the suggestions made by the reviewers. Those changes are highlighted within the manuscript. Please see below, in red, for a point-by-point response to the reviewers’ comments and concerns.
Reviewer #1
The authors of the article have done interesting work on determining the bioenergetic parameters of glycolysis and oxidative phosphorylation processes and their ratio in the studied cells. Nevertheless, the authors need to improve some points in the discussion of the results and conclusion.
Some comments:
Line 279. Misprints: foetal
Done, thanks.
Line 241. Only n fibroblast results in figure 2C highlighted the negative effect of mutation on the glycolysis pathway.
Thank. We have rewritten the sentence “Only fibroblast results in figure 2C highlighted the negative effect of mutation on the glycolysis pathway”
10% DMSO. DMSO itself is toxic, did you check viability/cell morphology after thawing the cells.
We fully agree with the reviewer: during the set-up of the freezing protocol, we verified that cell death after thawing never exceeded over 5%. We have edited the sentence accordingly.
How were cell lines treated to measure bioenergetics, such as substrate penetration into cells?
We have used the standardized protocol on the Agilent website for the following assays: ATP Rate Assay (https://www.agilent.com/cs/library/flyers/public/5991-9233EN_ATP_rate_kit_flyer.pdf), Cell Mito Stress Test (https://www.agilent.com/cs/library/usermanuals/public/XF_Cell_Mito_Stress_Test_Kit_User_Guide.pdf), Glycolytic Rate Assay (https://www.agilent.com/cs/library/usermanuals/public/103346-400.pdf), and Cell Energy Pheno-type Test (https://www.agilent.com/cs/library/usermanuals/public/XFp_Cell_Energy_Phenotype_Test_Kit_User_Guide.pdf).
Inhibitors used in the kits are membrane permeable, whereas the substrates (i.e., glucose, pyruvate and glutamine) are transported in the cells by the specific translocases (GLUT, Pyruvate carrier, and Glutamine carrier) (https://www.agilent.com/en/product/cell-analysis/real-time-cell-metabolic-analysis/xf-analyzers/seahorse-xf-hs-mini-analyzer-770502#howitworks).
Line 84. Consistently, the total ATP production of control fibroblasts and preadipocytes was 85% and 23% increased with respect to the FPLD2 fibroblasts and preadipocytes, respectively. It is more correct to write that total ATP production of FPLD2 fibroblasts and preadipocytes was decreased compared to control fibroblasts and preadipocytes.
We have rephrased the sentence as suggested, thank you.
Moreover, control and FPLD2 fibroblasts were about 150% and 65% larger than control and FPLD2 preadipocytes, respectively, due to a consequent rise in mitochondrial ATP synthesis. What exactly about 150% and 65% larger?
We thank the Reviewer for spotting this mistake. We have edited the sentence “Moreover, the total ATP production of control and FPLD2 fibroblasts was about 150% and 65% higher than control and FPLD2 preadipocytes, respectively, due to a consequent rise in mitochondrial ATP synthesis”.
Line106. The proton leak, which corresponds to the basal respiration subtracted from the respiration in the presence of oligomycin (minimal respiration), indicates the re-entry of H+ in the intermembrane space bypassing the F1FO-ATP synthase activity [13]. It is not quite clear from this sentence exactly how the proton leak is calculated. Please give a more exact definition (calculation) of the proton leak.
Thank you for pointing out this deficiency. We have rewritten the sentence as suggested.
“In general, basal respiration of cells is controlled strongly by ATP turnover and proton leak. The basal respiration inhibited by oligomycin (the specific inhibitor of ATP synthase) corresponds to the ATP turnover, whereas the residual activity or oligomycin-insensitive respiration is the proton leak. Therefore, the proton leak is the basal respiration subtracted from the respiration sensitive to oligomycin (ATP turnover), which indicates the re-entry of H+ in the intermembrane space bypassing the F1FO-ATP synthase activity [13].”
Line 127. 2.0 μM or 3.0 μM carbonyl-cyanide-4-(trifluoromethoxy) phenylhydrazone (FCCP) with fibroblasts and preadipocytes, respectively. Why such differences in concentration for different types of cells?
We have set the concentration after FCCP titration as required by Cell Mito Stress Test® by obtaining the results reported in the plots. 2.0 and 3.0 μM FCCP are the concentrations that induce the highest maximal respiration in fibroblasts and preadipocytes, respectively.
Please describe in more detail the definition of the parameters of spare respiratory capacity, ATP turnover.
We accepted the reviewer's suggestion.
Line 139. In FPLD2 and control fibroblasts and control preadipocytes, the coupling efficiency was 0.80±0.04, 0.76±0.03, 0.88±0.02 a.u., respectively, whereas FPLD2 preadipocytes had significantly lower value (0.67±0.04 a.u.). Is this data shown in the figures?
The coupling efficiency is an empirical value obtained indirectly from the mitochondrial respiration profile as ATP turnover/basal respiration ratio.
Line 163. ECAR впервые встречается в разделе результаты. Добавьте туда расшифровку аббревиатуры. ECAR appears in the results section for the first time. Add a transcription of the abbreviation there.
Thanks, done.
Fig. 2, 3 pmol/min, indicate by how many cells. Line 285 index of cell respiration (pmoL/min). The OCR and ECAR values were normalized to the total number of cells per well. It is more correct to indicate in the figures by how many cells, i.e. pmol/min/…cell.
The WAVE program of Seahorse has been set to compare the two different cell lines at 10K cells per well, but we have not modified the y-axis of Excel plots accordingly. We thank the Reviewer for spotting this mistake. Now, we have edited accordingly.
Line 185. stressed conditions. What kind of stressed conditions?
W have added in the text the meaning of “stressed conditions”. Oligomycin plus FCCP are stressor compounds used in the “Cell Energy Phenotype Test” to detect the response to induced energy demand by:
- a compensatory increase in the rate of glycolysis as the cells attempt to meet their energy demands via the glycolytic pathway when oligomycin blocks the mitochondrial ATP synthesis;
- in the presence of FCCP that drives oxygen consumption rates (OCR) higher as the mitochondria attempt to restore the mitochondrial membrane potential.
Line 231. We can assert that the FPLD2 alters negatively the physiological mitochondrial cycle of either fusion or fission which causes the deterioration of mitochondrial energy production. What the authors have in mind is a physiological mitochondrial cycle of either fusion or fission. In the FPLD2 cell line, ATP synthesis in the mitochondria is reduced. Is this event the cause of the impaired fusion or fission processes or does the impairment in fusion or fission lead to a decrease in ATP production specifically by the mitochondria?
We thank the referee for this thoughtful comment. This is a very interesting question! However, from the results of FPLD2 effects on cellular metabolism is not possible to understand the direct/indirect action on mitochondrial dynamics and morphology. We hope to study this intriguing relationship in the future work.
Line 265 Consistently, considering the cell metabolism results, we may argue that FPLD2 impaired mitochondrial homeostasis (biogenesis/mitophagy) and dynamics (fusion/fission) [21]. What exactly is happening with mitochondrial homeostasis (biogenesis/mitophagy) and dynamics (fusion/fission) in FPLD2 cells.
The FPLD2 effect on mitochondrial homeostasis and dynamics is still under-explored research topic. We have discussed the results in the “Discussion” section in relation to the possible effect of FPLD2 mutation in ageing and ageing in mitochondrial homeostasis and dynamics.
The conclusion turned out to have little to do with the main data of the article, as well as a discussion of the results.
We have taken into account the Reviewer's suggestion and we have revised the “Conclusion” section accordingly.

Reviewer 2 Report
The paper “Cellular metabolism and bioenergetic function in human fibroblasts and pre-adipocytes of type 2 familial partial lipodystrophy by Algieri et al it's an interesting paper but it’s quite difficult to read. Long sentences, not adequate terminolgy make generate confusion. English very often seems to be the result of a translated spoken language rather than the correct form of written English.
Deep english revision is required.
I will give below a major revision request of the Introduction and some examples of English phrases or constructions that can be misleading
Major revisions
Introduction
In the introduction the objective of the study is missing, the most part of the text being more in keeping with a clinical work than an experimental research study. The subject matter of the study, i.e., the experimental system used and the information that can be obtained from the study of fibroblasts and pre-adipocytes in FLPD2 is too general. It does not introduce the experimental design of the result section, moreover very well presented and discussed. Missing are references to what failure of cellular respiration pathways in the energy metabolism framework can be invoked as triggers of lipodystrophy in FLPD2.
The authors are asked to improve Introduction addressing it to the referee’s suggestions
Line 39
Unfold the OMIM acronym
Line 65
In the last sentence " The less-invasive biopsy-derived fibroblasts can give rise to other cells, including adipocytes, and this likewise happens for the preadipocytes" what does it mean? As first, it is low-invasive and not less-invasive, since the use of less implies a comparison. As a second, authors probably mean that " Fibroblast cultures obtained from low-invasive biopsies can generate other cell types, among them either adypocytes and pre-adipocytes?" or pre-adipocytes have a different origin. The sentence is very confusing.
Results
The experimental results are well described and the study design is clear. English also needs improvement in this section.
Paragraph 2.1
In the first sentence: “To characterize the cellular ATP production, namely ATP production rate, related to the conversion of glucose to lactate in the glycolytic pathway (glycoATP production rate) and to oxygen consumption in mitochondrial OXPHOS (mitoATP production rate)”, the verb is missing
line 93
The mutation (which mutation?: too generic word ) could modify ATP production, but the results showed that the ratio between mitoATP production rate vs glycoATP production rate was always > 1
I can suggest: Even if the mutation (which mutation?) could modify the ATP production, the ratio between mitoATP vs glycoATP production rates was always >1
line 97
Authors report “The cell respiration profile of FPLD2 and control fibroblasts and pre-adipocytes (Fig. 1C) was analyzed” with FPLD2 they want to refer to FPLD2 fibroblasts and to FPLD2 pre-adypocytes as can be envisaged from 1C?. If yes, please use an expression that allow them to be distinguished.
Line 100
Unfold the two acronyms OCR and FCCP when given as firts time in the text and not in figure caption
line 149
In the sentence “We noticed for this parameter a similar trend similar……. Eliminate similar before trend
Paragraph 2.2
Line 163
Unfold the acronym ECAR when given as firts time in the text and not in figure caption
Discussion
Line 230
In the sentence “ On the other hand cellular energy metabolism influences mitochondrial dynamics morphology in turn, probably it would be better ……………dynamics of mitochondrial morphology
Line 241
n to be canceled
line 249
The depicted quiescent profile instead of The quiescent profile depicted
The sentence at line 235-238 is long-winded.
Author Response
Thank you for giving us the opportunity to submit a revised draft of the manuscript “Cellular metabolism and bioenergetic function in human fi-broblasts and pre-adipocytes of type 2 familial partial lipo-dystrophy” for publication in the Int. J. Mol. Sci. We appreciate the time and effort that you and the reviewers dedicated to providing feedback on our manuscript and are grateful for the insightful comments on and valuable improvements to our paper. We have incorporated most of the suggestions made by the reviewers. Those changes are highlighted within the manuscript. Please see below, in red, for a point-by-point response to the reviewers’ comments and concerns.
Reviewer #2
Comments and Suggestions for Authors
The paper “Cellular metabolism and bioenergetic function in human fibroblasts and pre-adipocytes of type 2 familial partial lipodystrophy by Algieri et al it's an interesting paper but it’s quite difficult to read. Long sentences, not adequate terminolgy make generate confusion. English very often seems to be the result of a translated spoken language rather than the correct form of written English.
Deep english revision is required.
We are pleased that the Reviewer considers our work of value. According to the reviewer’s comment, we have checked the manuscript to improve the written English form.
I will give below a major revision request of the Introduction and some examples of English phrases or constructions that can be misleading
Major revisions
Introduction
In the introduction the objective of the study is missing, the most part of the text being more in keeping with a clinical work than an experimental research study. The subject matter of the study, i.e., the experimental system used and the information that can be obtained from the study of fibroblasts and pre-adipocytes in FLPD2 is too general. It does not introduce the experimental design of the result section, moreover very well presented and discussed. Missing are references to what failure of cellular respiration pathways in the energy metabolism framework can be invoked as triggers of lipodystrophy in FLPD2.
The authors are asked to improve Introduction addressing it to the referee’s suggestions
There was an editorial misunderstanding and the paper was initially written as a “Communication” and transformed into “Article”. This is the reason for the deficiencies in the introduction. We took the reviewer's advice and tried to improve it.
Line 39
Unfold the OMIM acronym
Done, thanks.
Line 65
In the last sentence " The less-invasive biopsy-derived fibroblasts can give rise to other cells, including adipocytes, and this likewise happens for the preadipocytes" what does it mean? As first, it is low-invasive and not less-invasive, since the use of less implies a comparison. As a second, authors probably mean that " Fibroblast cultures obtained from low-invasive biopsies can generate other cell types, among them either adypocytes and pre-adipocytes?" or pre-adipocytes have a different origin. The sentence is very confusing.
We thank the Reviewer for spotting this mistake. We have rephrased the sentence as suggested, thank you.
“Fibroblast cultures obtained from low-invasive biopsies can generate other cell types, among them either adypocytes and pre-adipocytes”
Results
The experimental results are well described and the study design is clear. English also needs improvement in this section.
Paragraph 2.1
In the first sentence: “To characterize the cellular ATP production, namely ATP production rate, related to the conversion of glucose to lactate in the glycolytic pathway (glycoATP production rate) and to oxygen consumption in mitochondrial OXPHOS (mitoATP production rate)”, the verb is missing
Thank you for pointing out this deficiency. We have rewritten the sentence.
line 93
The mutation (which mutation?: too generic word ) could modify ATP production, but the results showed that the ratio between mitoATP production rate vs glycoATP production rate was always > 1
I can suggest: Even if the mutation (which mutation?) could modify the ATP production, the ratio between mitoATP vs glycoATP production rates was always >1
We have taken into account the Reviewer's suggestion. Thanks.
“Even if FPLD2, which is caused by an autosomal dominant mutation in the LMNA gene, could modify ATP production, the ratio between mitoATP vs glycoATP production rates was always > 1”
line 97
Authors report “The cell respiration profile of FPLD2 and control fibroblasts and pre-adipocytes (Fig. 1C) was analyzed” with FPLD2 they want to refer to FPLD2 fibroblasts and to FPLD2 pre-adypocytes as can be envisaged from 1C?. If yes, please use an expression that allow them to be distinguished.
Done, thanks
Line 100
Unfold the two acronyms OCR and FCCP when given as firts time in the text and not in figure caption
Done, thanks.
line 149
In the sentence “We noticed for this parameter a similar trend similar……. Eliminate similar before trend
Done, thanks.
Paragraph 2.2
Line 163
Unfold the acronym ECAR when given as firts time in the text and not in figure caption
Done, thanks.
Discussion
Line 230
In the sentence “On the other hand cellular energy metabolism influences mitochondrial dynamics morphology in turn, probably it would be better ……………dynamics of mitochondrial morphology
Thank you for this suggestion. We have changed the sentence accordingly.
Line 241
n to be canceled
Correct, thanks.
line 249
The depicted quiescent profile instead of The quiescent profile depicted
Done, thanks.
The sentence at line 235-238 is long-winded.
We have rephrased the sentence as suggested, thank you.

Round 2
Reviewer 1 Report
Thanks to the authors for the quick and detailed answers to the comments.
Reviewer 2 Report
Authors made efforts to improve the paper content. The english language improved too. As stated in the previous review the paper is interesting and is now ready for publiucation in IJMS